# The Influence of Food Intake and Blood Glucose on Postprandial Sleepiness and Work Productivity: A Scoping Review

**DOI:** 10.3390/nu17203217

**Published:** 2025-10-14

**Authors:** Hisashi Kaneda, Itsuki Kageyama, Yoshiyuki Kobayashi, Kota Kodama

**Affiliations:** Medical Data Science Lab, Hoshi University, Tokyo 142-0063, Japan

**Keywords:** postprandial hypoglycemia, sleepiness, worker, scoping review

## Abstract

**Background:** Occupational accidents, injuries, and illnesses are serious problems for organizations. Workplace sleepiness is a major issue that affects occupational safety and productivity. Workplace sleepiness is influenced by sleep, diet, and blood glucose levels, but the causal relationship is unclear. This scoping review aimed to investigate the factors affecting work productivity, with a particular focus on the impact of sleepiness caused by food intake and blood glucose level on productivity. **Methods:** PubMed, and Web of Science were used to search terms, such as “workplace,” “sleepiness or postprandial hypoglycemia,” “productivity,” and “measurement.” The following studies were included: (1) those with working hours evaluations; (2) that excluded patients with diabetes, heart diseases, or other diseases; (3) that excluded patients with mental illness; (4) that did not limit the evaluation of sleepiness at work to sleep only; (5) with publications after 2014; and (6) that were research articles. **Results:** The search yielded 521 articles. Nine papers met the inclusion criteria. Six studies assessed blood glucose levels, six assessed sleepiness, and one simultaneously assessed blood glucose and sleepiness. The Karolinska Sleepiness Scale (KSS) is the most frequently used sleep measure. Most studies have assessed the effects of controlled diets. Although some studies have used continuous glucometers, their evaluation of blood glucose levels has been limited. The extracted literature also included mild exercise and control of environmental illumination as a countermeasure against sleepiness. **Conclusions:** Although few studies have analyzed the causes and countermeasures of sleepiness in the workplace, sleepiness affects work productivity, diet affects sleepiness, and several methods for suppressing sleepiness have been researched. However, a few studies have directly evaluated the effects of blood glucose fluctuations and sleepiness on work productivity. These results suggest that further research into the relationship between sleepiness at work and related biological signals and blood glucose fluctuations will be important in understanding the causes, as it will form the basis for measures to improve work productivity.

## 1. Background

Worker productivity is crucial in advancing ecologically sustainable economic development and optimizing resource utilization. By enhancing worker productivity, nations lay the foundation for economic prosperity and significantly contribute to a sustainable future [1]. Companies are taking various approaches to improve productivity, such as improving throughput by improving production systems [2], strategies to grow the company, stimulate and retain employee potential [3], and the introduction of industrial robots [4]. A study of a panel of 25 OECD economies also shows that real wages are positively correlated with worker productivity [5]. However, productivity challenges remain. While automation technologies are helping to improve productivity, the potential for automation of non-routine work activities remains limited, and tasks such as people management and development, decision-making, planning, and creative tasks remain less amenable to automation [6]. Furthermore, productivity is positively correlated with innovation not just in the manufacturing process, but in both small and large companies, formal and informal companies, and developed and emerging countries [7]. Improving productivity is thus an important issue, and a company’s productivity depends in part on the workers productivity. Research focusing on worker productivity has been conducted from multiple angles. One study of call center agents found that as working time increases, output per hour decreases [8]. A study using the UKHLS (also known as ‘Understanding Society’, a large-scale national household panel survey that covers a representative sample of UK households administered from 2009) found: regarding the expansion of working from home since COVID-19, on average, workers report being at least as productive as before the pandemic began. However, this average mask substantial heterogeneity, which is linked to job quality, gender, the presence of children, and ease of working from home [9]. A study on hybrid work conducted between 2021 and 2022 among 1,612 employees at a Chinese technology company showed that a hybrid schedule of working from home two days a week did not hurt performance [10]. In terms of the working environment, a study in China found that a 10-unit increase in air pollution index led to a significant decrease in labor productivity by 4% [11].

While these organizational and environmental factors are significant, it is also crucial to focus on the productivity of individual workers. As automation handles more routine work, the value of human labor increasingly shifts to cognitive tasks where an individual’s physiological state—governing factors like concentration and alertness—plays a decisive role in their performance [6]. Therefore, understanding the immediate factors that impact a worker’s daily capacity, such as sleepiness, becomes a central issue for improving modern workplace productivity. Both sleep quality and quantity have a direct impact on workplace performance and sleepiness [12]. Among older workers, blue-collar workers experienced lower sleep quality, productivity, and cognitive function compared to white-collar workers, suggesting a need to promote healthy sleep habits [13]. In industries that require continuous 24 h operation, the introduction of limited wake shift work (LWSW) has attracted attention; LWSW schedules have been linked to improvements in sleep, wakefulness, and performance. These advantages are particularly apparent in the following scenarios: (i) shorter time at work, (ii) more frequent rest breaks, (iii) shifts that start and end at the same time every 24 h, and (iv) work shifts commencing during the daytime (as opposed to at night) [14]. Furthermore, workers who are, on average, sleep-deprived exhibit low safety compliance [15], indicating that companies should design labor management systems that prioritize workers’ sleep quality and quantity [16]. However, to the best of our knowledge, a causal relationship between sleepiness and work is not fully understood. In general, sleep and eating behavior are widely recognized as bidirectionally related, and different meal types (including foods and beverages) have been reported to affect sleepiness and alertness. In particular, fluctuations in blood glucose levels after eating and drinking are considered causes of sleepiness [17,18]. Nevertheless, despite many workers having time to eat and drink during work hours, comprehensive discussions that jointly examine workers’ diet and glycemic variability, sleep, and work productivity (performance) remain scarce [19,20].

The purpose of this scoping review is to focus on work productivity and the relationship between sleepiness due to meal and blood glucose fluctuations that affect work productivity. In addition, we investigated methods to eliminate sleepiness during work. This study focuses on the effects of food, focusing on “sleepiness”. However, “drowsiness” caused by medication side effects was excluded from the study. Also, by excluding evaluations of patients with diabetes, studies on fatigue only, and studies conducted outside of work, we aimed to gain a comprehensive understanding of the factors that affect productivity in the workplace.

## 2. Methods

### 2.1. Scoping Review and Search Strategy

To effectively chart the scope, nature, and current gaps in the research, we adopted a scoping review methodology. This approach is designed to provide a broad overview of a research area by summarizing its core concepts and available evidence. The entire review process was guided by the foundational framework for scoping reviews proposed by Arksey and O’Malley and Levac et al. [21,22,23], and our reporting follows the Preferred Reporting Items for Systematic Reviews and Meta-Analyses Extension for Scoping Reviews (PRISMA-ScR) checklist.

To build a robust research foundation for this review, we performed a systematic search of two key scientific databases: PubMed and Web of Science. The selection of PubMed ensures comprehensive coverage of high-quality, peer-reviewed literature in the medical and life sciences. This was complemented by the use of Web of Science to capture a broader range of high-impact, international research. The search was executed using keywords including “workplace,” “sleepiness or postprandial hypoglycemia,” “productivity,” and “measurement,” as specified in Appendix A. To complement the initial search, we conducted a thorough targeted search of the gray literature using Google Scholar and Web of Science to identify non-indexed studies, including unpublished trial data, articles or dissertations, and conference proceedings. The reference lists of relevant reviews and included studies were also manually searched for relevant studies that were not captured in the initial search; however, these were not included in the final list of reviewed studies.

### 2.2. Eligibility Criteria

Population, concept, and situation approaches were used to facilitate the development of eligibility criteria and standardize screening approaches (Table 1). For inclusion in this review, studies related to glycemic fluctuations and sleepiness in workers at work were needed because of factors, such as meal. The work environment was not limited to a specific environment, and experiments simulating a work environment were included. There were also no restrictions on the type of food consumed.

The inclusion criteria were as follows: studies that (1) were related to workers’ workplace glycemic variability and sleepiness attributable to factors such as meals; (2) included evaluations on hours during or simulating working hours; (3) did not limit sleepiness at work to sleep only; (4) included publications after 2014: labor management laws have been revised in many countries in recent years, so we conducted searches over a period of approximately ten years to compile results that can be used to make recommendations for the future. Therefore, this study period was from 2014 to 2024 (7 October 2024); (5) were peer-reviewed papers, experimental studies, and observational studies that described the implementation of specific interventions; and (6) contained quantitative and/or qualitative data and had well-defined outcomes (outcomes were defined as the inclusion of validated measures of sleepiness indicators or blood glucose levels).

To exclude influences other than work and meals and to limit the study to actual workers, we applied the following exclusion criteria: (1) studies including patients with diabetes, heart disease, or other physical diseases; (2) studies including patients with mental illness; (3) studies that did not evaluate the relationship between food intake, sleepiness, and productivity; (4) studies involving students or those conducted in sports environments; and (5) studies that included data from non-working hours.

### 2.3. Data Extraction and Coding

An initial search was conducted in the selected databases using the selected search terms. For example, search terms for the PubMed database were used to generate initial hits, yielding 313 results. Subsequently, the search period was narrowed to 1 January 2014, to 7 October 2024, yielding 200 hits, and further narrowed to English-language articles, yielding 193 hits. All database search results were collected into a single file. An Excel 2019 spreadsheet was used to store, organize, and code the data. Duplicates were then removed, and the titles and abstracts of all remaining articles were reviewed.

### 2.4. Study Selection

Two reviewers screened all titles and abstracts and assessed their eligibility for inclusion. Articles were excluded based on the exclusion and inclusion criteria (remaining *n* = 45). All papers containing the search terms in the title and/or abstract were included in this stage. If it was unclear whether an article should be included or whether to include or exclude it if an abstract was not available, the decision to include or exclude was made after reading the entire article in question and discussing it with all reviewers.

In the next stage, the full text of all selected articles was read and reviewed by two reviewers. Once the final set of articles was selected based on their relevance to the research question, they were coded and categorized using Excel 2019 (remaining *n* = 9). All reviewers were involved in the coding process.

## 3. Results

The literature search revealed 521 citations and all papers were reviewed (Figure 1). Only nine studies met the eligibility criteria. The nine studies contained six blood glucose measurement evaluations and six questionnaires sleepiness assessment (Table 2 and Table 3). The experimental studies by Liviya et al. [24], Wehrens et al. [25], and Wennberg et al. [26] included both aspects. Continuous glucose meters (CGMs) were used to measure blood glucose levels in two studies and one study used fixed-point blood sampling. Four studies used the Karolinska Sleepiness Scale (KSS), Epworth Sleepiness Scale (ESS), and visual analog scale (VAS) to evaluate fatigue severity.

The 2014 study by Liviya et al. [24] investigated the prevalence of excessive daytime sleepiness (EDS) among 707 Australian workers. Using the Epworth Sleepiness Scale (ESS), they found that factors such as older age, higher BMI, unhealthy dietary habits, and poor mental health were linked to higher EDS scores. The study concluded that workplace wellness programs focusing on diet and weight management could potentially reduce EDS, thereby improving both productivity and mental well-being. However, a key limitation of this research, for the purposes of our review, was its lack of assessment of blood glucose variability or the direct impact of individual meals on sleepiness, as diet was only assessed broadly through questionnaires.

In the study by Wehrens et al. [25], the effect of meal timing on circadian rhythms was examined in a controlled laboratory setting. The researchers subjected participants to a 13-day protocol where meals were delayed by five hours. While this shift in meal timing did not alter subjective feelings of sleepiness or hunger, it produced significant changes in metabolic markers. Specifically, the circadian rhythm of plasma glucose was delayed by approximately 5.7 h, and the average glucose concentration also decreased. These findings suggest that meal timing is a key factor in regulating peripheral circadian rhythms, such as glucose metabolism, even when it does not affect subjective sleepiness.

The study by Wennberg et al. [26] examined if brief, regular walking breaks could mitigate fatigue in sedentary, overweight adults. While the intervention—three-minute walks each hour—successfully reduced participants’ subjective feelings of fatigue, it did not lead to any measurable changes in postprandial blood glucose levels. The findings suggest that even light physical activity can be an effective strategy for combating workplace fatigue, though this study could not establish a direct link between this effect and blood glucose regulation.

Brocklebank et al. [27] investigated the acute effects of incorporating short periods of standing or light walking on postprandial blood glucose levels in sedentary office workers. Under controlled conditions that accounted for participants’ meals on the previous evening and morning of the trial, they used a continuous glucose monitor (CGM iPro2) to compare uninterrupted sitting with two active break protocols: two minutes of standing or two minutes of light walking, both performed every 20 min. The results showed that these brief activity breaks improved glucose regulation, with the most pronounced benefits seen in overweight male participants.

The pilot study by Han et al. [28] assessed the impact of using a pedal desk on the metabolic responses of sedentary office workers. Participants were tested in two conditions: sitting normally for 120 min versus pedaling continuously for the same duration. The key finding was that while continuous pedaling did not negatively affect work-related skills, it significantly lowered postprandial insulin concentrations. This suggests that incorporating light, continuous activity via a pedal desk could be a viable strategy for improving metabolic health in a workplace setting without compromising productivity.

Ferreira et al. [29] conducted a study to determine if various biomarkers could predict work productivity. Unlike studies focusing on dynamic glucose changes, they used a single fasting blood glucose measurement alongside other markers like thyroid-stimulating hormone (TSH). Their analysis revealed that lower work productivity was associated with higher levels of both fasting blood glucose and TSH, suggesting that even static metabolic and hormonal markers can be linked to performance outcome.

Three studies evaluated the relationship between sleepiness at work and controlled eating. The KSS was used to assess all types of sleepiness. These three studies explored different approaches to eliminate sleepiness.

Askaripoor et al. [30] tested the hypothesis that specific light conditions could counteract the common post-lunch dip in alertness. In their study, participants were exposed to one of several types of light after their midday meal, including blue-rich white light (BWL) and red-saturated white light (RWL). The results, confirmed with electroencephalography (EEG), showed that both BWL and RWL were effective in enhancing alertness when compared to normal or dim lighting conditions.

Kowalsky et al. [31] conducted a randomized crossover trial to assess whether using a sit-stand desk could alleviate sleepiness during a simulated 8 h workday. While participants in both the prolonged sitting (SIT) and sit-stand (SIT-STAND) conditions reported increased sleepiness over the day, the increase was significantly less pronounced in the SIT-STAND group. This suggests that regularly alternating between sitting and standing is an effective strategy for mitigating the build-up of workday sleepiness, with the authors noting the effect was particularly significant after lunch.

In a subsequent study, Kowalsky et al. [32] evaluated the effect of hourly resistance exercise breaks (REX) compared to uninterrupted sitting (SIT). While the exercise breaks did not produce a statistically significant overall reduction in sleepiness, they did lead to a significant decrease in mental fatigue by the end of the four-hour session. This suggests that incorporating brief resistance exercises into the workday is a highly acceptable and effective strategy for combating mental fatigue, even if its impact on general sleepiness is less pronounced.

To compare studies that examined the relationship between light exercise and blood glucose levels, the dietary and exercise conditions of Wennberg et al. [26], Brocklebank, LA et al. [27], and Han et al. [28] are summarized in Table 4. The study by Brocklebank, LA et al. [27] found a significant difference in blood glucose fluctuations with or without light exercise, but the remaining two studies found no significant difference.

To compare studies that examined the relationship between light exercise and sleepiness, the dietary and exercise conditions of Kowalsky et al. (2018) [31] and Kowalsky et al. (2021) [32] are summarized in Table 5. Neither study found a clear result that exercise reduced sleepiness. Although the included studies aimed to simulate working hours, their experimental protocols deviated from typical workplace conditions. Specifically, the prescribed fasting periods and exercise interventions do not reflect standard practices in a real-world occupational setting, which may limit the generalizability of the findings.

## 4. Discussion

This review clearly shows that although sleepiness at work is a notable issue affecting work productivity and employee health, there is a lack of literature evaluating the association between sleepiness at work and diet or blood glucose fluctuations. In the cross-sectional study by Liviya et al. [24], unhealthy dietary habits, higher BMI, and poorer mental health were associated with EDS, suggesting that adverse lifestyle patterns affect sleep status. Individuals with EDS are prone to inattention and reduced work efficiency, with a potential increase in errors and accidents in the workplace. Accordingly, improving diet quality and managing body weight may enhance sleep quality and daytime alertness, with the potential to improve productivity. By contrast, Wehrens et al. [25] showed that meal timing is a key factor that entrains peripheral metabolic rhythms (peripheral clocks) independently of the sleep–wake rhythm. When meal times are delayed, the circadian rhythm of blood glucose is likewise delayed. Thus, for productivity it is important not only what is eaten but also when it is eaten; particularly for night-shift workers, strategic meal timing may be effective for both health management and operational efficiency. Furthermore, Wennberg et al. [26] demonstrated that brief walking breaks every 30 min significantly attenuate the accumulation of fatigue, providing physiological evidence that fatigue-related performance decrements can be mitigated. By contrast, no between-condition differences were detected for interstitial glucose (CGM) or insulin, and within the acute intervention/observation window of this study, neither improvements in glycemic variability nor in cognitive test performance were observed. Although some studies (e.g., Brocklebank et al. [27]) have reported changes in postprandial glycemic responses with light-intensity activity, interpreting these discrepant findings is challenging due to methodological inconsistencies across studies. A primary issue is the variability in exercise interventions; for instance, protocols ranged from two-minute walking breaks every 20 min to continuous pedaling, making it difficult to establish a clear dose–response relationship. Compounding this, the dietary protocols were also highly heterogeneous, with meals varying substantially in energy content, macronutrient composition, and timing. This lack of standardization in both exercise and dietary controls complicates direct comparisons and may confound the observed outcomes. Nevertheless, the fact that brief activity breaks suppress fatigue and help maintain concentration is of high practical value and represents a promising strategy to enhance productivity by sustaining attention and vigor. In conclusion, sleep, dietary habits (including timing), and activity patterns in the workplace are closely interrelated with health and productivity. Improving lifestyle and work practices—ensuring adequate sleep and daytime alertness, adopting nutritionally balanced diets with appropriate timing, and incorporating frequent short breaks or light activity during work—can be expected to improve sleep quality, reduce fatigue, and support glycemic control, thereby ultimately enhancing productivity. The assessment of sleepiness during work has been conducted in certain occupations and environments. Labor productivity decreases because of sleepiness during work. Extensive evaluations have been conducted among nurses, offshore workers in the oil and gas industry, drivers, and pilots. Shift workers are more tired or drowsier than non-shift workers and night-shifts are more fatigued or sleepier than day-shift workers [33,34,35]. However, some studies have shown that subjective and objective sleepiness is contradictory, and the establishment of evaluation methods in daily intervention trials is challenging [36,37,38]. Similarly to Askaripoor et al. [30], the results of evaluating the difference in sleepiness depending on the color of light confirmed the ability of BWL to reduce sleepiness [39]. Drivers are also exposed to street lighting, which affects mood and sleepiness [40]. Therefore, the impact of light on sleepiness is significant, warranting future detailed research is warranted. All participants considered sleep duration and did not refer to eating or drinking during their working hours.

Many studies have been conducted on the assessment of sleepiness during work via questionnaires or on its correlation with sleep duration and quality of the previous day. However, it is even more important to understand the bio-signals that cause sleepiness. Recently, the widespread use of CGM has made it easier to measure blood glucose levels, which is not only used in patients with diabetes [41]. The research targets are the “Screening Tool for the Early Detection of Abnormal Glucose Regulation” and “Nutritional Behavior” related to diabetes prevention, “Stress,” and “Optimization of Athletic Performance”. A notable 2024 study by Yao et al. [42], which analyzed a large dataset of over 11,000 diets from 789 non-diabetic individuals using CGM, provided several key insights into lifestyle behaviors and postprandial glucose control. Their findings highlighted that diets high in carbohydrates (especially refined grains) and fried foods, but low in protein, were strongly linked to elevated postprandial glucose. Conversely, physical activity, particularly light-intensity movement after meals, as well as adequate sleep, were associated with significantly lower glucose responses. The study also observed a diurnal pattern, with postprandial glucose levels being lowest in the morning and peaking in the afternoon. These detailed findings suggest that applying similar CGM-based evaluations in a workplace context could offer powerful insights into how blood glucose fluctuations impact employee well-being and productivity.

Following the evolution of CGM, the possibility of noninvasive blood glucose measurement has become apparent. It is now possible for individuals other than patients with diabetes to continuously monitor blood glucose and CGM is expected to be used for a variety of applications [41]. It is expected that this will contribute to understanding the effects and causes of sleepiness in eating and drinking during work, and contribute to improving the decline in labor productivity due to sleepiness.

Building on recent technological advances, future applied workplace research should incorporate the assessment of key potential confounders. Individual chronotype (e.g., morningness–eveningness), job characteristics (cognitive vs. physical workload), and caffeine consumption each influence both sleepiness and glycemic responses. For chronotype, individuals with stronger eveningness or greater social jetlag tend to exhibit lower daytime alertness, higher fasting glucose and HbA1c levels, and an elevated risk of type 2 diabetes [43]. Regarding job characteristics, prolonged cognitively demanding, predominantly sedentary work increases mental fatigue and sleepiness. In contrast, interrupting sedentary time with short bouts of light walking reduces postprandial glycemic and insulinemic responses, as shown by intervention trials and reviews. Moreover, physically active tasks enhance skeletal muscle glucose uptake and improve insulin sensitivity; for example, aerobic exercise augments mitochondrial function and insulin receptor activity, and with continued practice, lowers fasting glucose and HbA1c levels. By contrast, seated cognitive work expends little energy and predisposes workers to postprandial hyperglycemia. Experimentally, brief stair-climbing after meals significantly attenuates the rise in postprandial glucose, indicating that modest activity breaks blunt glycemic spikes and supports the utility of exercise interventions within cognitively oriented, sedentary workplaces [44]. These observations are consistent with, and extend, prior findings by Wennberg et al. [26] and Brocklebank et al. [27]. Caffeine use is also common among workers (e.g., coffee, energy drinks). Through central adenosine-receptor antagonism, caffeine increases wakefulness and reduces sleepiness; however, it acutely decreases insulin sensitivity and can transiently shift glycemia upward [45].

Taken together, these mechanisms and empirical findings suggest that chronotype, job characteristics, and caffeine intake can serve as significant confounders when assessing postprandial sleepiness and glycemic trajectories. Appropriately controlling these variables is essential for isolating the specific impact of dietary factors on productivity and for designing more targeted and effective workplace interventions.

However, this review has several limitations that must be acknowledged. First, a significant limitation within the existing literature is that most of the included studies had small sample sizes, which restricts their statistical power and the generalizability of the findings. Second, very few studies simultaneously assessed the key variables of sleepiness, blood glucose, and productivity, making it difficult to establish causal relationships. Third, the potential influence of sex-specific hormonal factors was not considered in the reviewed literature. Hormonal fluctuations related to the menstrual cycle or menopause can modulate glucose metabolism and sleep quality, yet these variables are rarely controlled for. Future research should incorporate sex-stratified analyses to clarify these interactions. Methodologically, while our search included multiple databases and gray literature, other databases such as the Cochrane Library and Scopus were not searched, and a formal quality assessment of the included studies was not performed. Finally, the exclusion of studies involving participants with diagnosed conditions, while necessary for this review’s focus, may limit the applicability of our findings to real-world workforces where chronic illnesses are common.

## 5. Conclusions

Although few studies have analyzed the causes and countermeasures of sleepiness in the workplace, sleepiness affects work productivity, diet affects sleepiness, and several methods for suppressing sleepiness have been researched. However, a few studies have directly evaluated the effects of blood glucose fluctuations and sleepiness on work productivity. These results suggest that further research into the relationship between sleepiness at work and related biological signals and blood glucose fluctuations will be important in understanding the causes, as it will form the basis for measures to improve work productivity.

## Figures and Tables

**Figure 1 nutrients-17-03217-f001:**
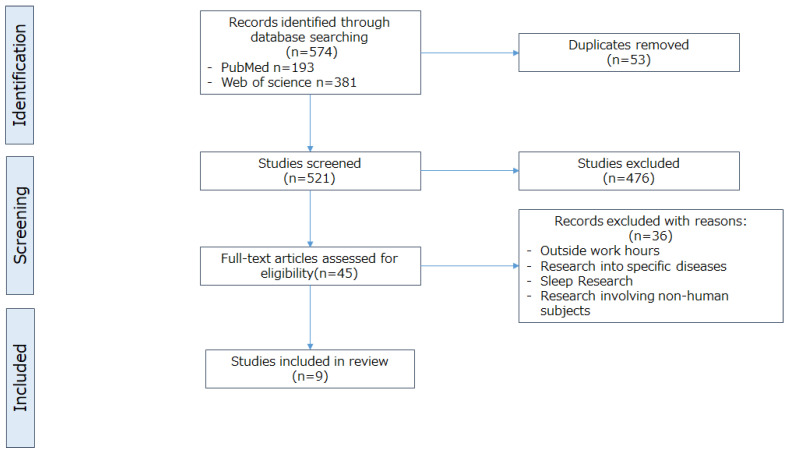
PRISMA flow diagram and list of excluded full-text studies with reasons.

**Table 1 nutrients-17-03217-t001:** Inclusion and exclusion criteria.

	Population	Context	Concept
Included	Participants who mimic workers or working conditions	- Changes in sleepiness after meals - Changes in productivity metrics (objective or subjective) linked to post-meal states	At work
Excluded	Participants included patients with diagnosed conditions such as diabetes, heart disease, and psychiatric disorders (This does not include people who have risk factors for certain diseases, such as obesity or a high BMI, but who have not yet been diagnosed)	- Sleepiness that is unrelated to food intake - Productivity unrelated to drowsiness	Non-working hours, students’ study environment, and sports environment

**Table 2 nutrients-17-03217-t002:** Characteristics of the included studies.

Index	Author/Publication Year	Type of Study	Blood Glucose Measurement	Sleepiness Assessment	Dietary Control	Sample Size	Number of Females/ Males	Age (Range)
1	Liviya, Winda et al. (2014) [24]	Observational Study	Less than once a day	Epworth Sleepiness Scale	Uncontrolled. Participants’ eating habits were investigated *	707	Female 424 Male 283	40.2 ± 10.4
2	Wehrens, Sophie M. T. et al. (2017) [25]	Crossover Trial	Multiple blood glucose sampling	Karolinska Sleepiness Scale	Controlled	10	Male 10	24 ± 6
3	Wennberg P et al. (2016) [26]	Randomized Crossover Trial	Continuous blood glucose monitoring	VAS to evaluate fatigue severity (VAS-F)	Controlled	19	Female 8 Male 11	59.7 ± 8.1
4	Brocklebank, LA et al. (2017) [27]	Randomized Crossover Trial	Continuous blood glucose monitoring	Not measured *	Controlled	17	Female 9 Male 8	52.4 ± 5.1
5	Han HO et al. (2018) [28]	Crossover Trial	Less than once a day	Not measured *	Controlled	12	Female 6 Male 6	42.5 ± 21.5
6	Ferreira AI et al. (2021) [29]	Observational Study	Less than once a day	Not measured *	Uncontrolled *	180	Female 137 Male 43	41.22 (No range specified, SD = 13.58)
7	Askaripoor T et al. (2019) [30]	Crossover Trial	Not measured *	Karolinska Sleepiness Scale	Uncontrolled	20	Male 20	27.65 ± 3.65
8	Kowalsky RJ et al. (2018) [31]	Randomized Crossover Trial	Not measured *	Karolinska Sleepiness Scale	Controlled	25	Female 9 Male 16	42.5 ± 22.5
9	Kowalsky RJ et al. (2021) [32]	Randomized Crossover Trial	Not measured *	Karolinska Sleepiness Scale	Controlled	14	Female 12 Male 3	53.4 ± 9.5

* Shaded areas are not applicable.

**Table 3 nutrients-17-03217-t003:** Summary of results of each study.

Index	Author/Publication Year	Summary
1	Liviya, Winda et al. (2014) [24]	Characteristics associated with EDS and higher Epworth Sleepiness Scale scores were older age, higher BMI, poor diet, and poor mental health. However, it did not investigate direct links between sleepiness and blood glucose or individual meals.
2	Wehrens, Sophie M. T. et al. (2017) [25]	In an experiment simulating shift workers and long-distance commuters, this study assessed rhythmic fluctuations in sleepiness based on meal timing. It found that delaying meals significantly altered peripheral circadian rhythms, such as the plasma glucose cycle, even though it did not affect participants’ subjective feelings of hunger or sleepiness.
3	Wennberg P et al. (2016) [26]	While regular, light-intensity walking breaks were effective at reducing subjective fatigue in sedentary adults, the intervention did not produce any significant changes in postprandial blood glucose levels.
4	Brocklebank, LA et al. (2017) [27]	The study demonstrated that interrupting prolonged periods of sitting with brief, light-intensity walking breaks significantly reduces postprandial glucose levels in middle-aged adults without metabolic disorders.
5	Han HO et al. (2018) [28]	Using a pedal desk for light physical activity during work did not alter plasma glucose concentrations but did lead to a significant reduction in postprandial insulin levels, suggesting improved insulin sensitivity.
6	Ferreira AI et al. (2021) [29]	This research linked various biomarkers to presenteeism (working while sick), finding that higher levels of thyroid-stimulating hormone and blood glucose were associated with lower employee job performance.
7	Askaripoor T et al. (2019) [30]	This study showed that exposure to specific types of light, particularly blue-rich (BWL) and red-saturated (RWL) white light, after lunch can effectively enhance alertness and counteract the post-lunch dip in performance.
8	Kowalsky RJ et al. (2018) [31]	This study suggests that using a desk that allows for both sitting and standing during simulated work hours may help reduce symptoms such as drowsiness, physical fatigue, and discomfort.
9	Kowalsky RJ et al. (2021) [32]	In this study, workers completed two 4 h conditions in a random order: one of prolonged sitting (SIT) and another of sitting interrupted by hourly resistance exercise breaks (REX). Although ratings for discomfort, fatigue, and sleepiness were typically lower during the REX condition compared to SIT, the overall outcomes were not significantly different between the two conditions. However, a significant decrease in mental fatigue was observed during the 4th hour in favor of the REX condition.

**Table 4 nutrients-17-03217-t004:** Characteristics of studies that examined the relationship between light exercise and blood glucose levels.

Index	Author	Meal	Exercise	Effect of Post-Meal Exercise on Blood Glucose Fluctuations
3	Wennberg P et al. [26]	The evening before the trial, participants consumed a standardized meal pack. On the test day, a high-fat, high-carbohydrate liquid meal (50g fat, 75g carb) was provided two hours into the session.	A crossover design compared 5 h of uninterrupted sitting against a condition where sitting was broken up by 3 min bouts of light-intensity walking every 30 min.	The study found no statistically significant differences in interstitial glucose levels (total AUC or net iAUC) between the active and sedentary conditions.
4	Brocklebank, LA et al. [27]	Participants ate a standardized meal at home the evening prior to the study and fasted overnight. On the morning of the trial, they consumed two standardized test drinks before the session began.	The study compared two five-hour conditions: one group remained seated continuously, while the other group interrupted their sitting with two-minute, light-intensity walks every 20 min.	Interrupting sitting with light-intensity walking resulted in a 55.5% lower five-hour incremental area under the curve (iAUC) for interstitial glucose compared to the uninterrupted sitting condition.
5	Han HO et al. [28]	Before the start of the experiment, the subjects were given a standard liquid meal containing 75 g of carbohydrates and 50 g of fat, which they consumed over a period of 10 min.	The study compared two 2 h conditions: (1) performing computer tasks while seated (STD). (2) performing the same tasks while pedaling at a light intensity using a pedal desk (PD).	There were no significant differences between the pedaling and standard sitting conditions in terms of glucose AUC, peak glucose concentration, or plasma glucose at any time point.

**Table 5 nutrients-17-03217-t005:** Characteristics of studies that examined the relationship between light exercise and sleepiness.

Index	Author	Meal	Exercise	Effect of Post-Meal Exercise on Sleepiness
8	Kowalsky RJ et al. [31]	After a 12 h fast, participants consumed standardized breakfast and lunch meals, each providing 30% of their daily caloric needs. The simulated workday was divided into two 3 h 40 min test sessions, one following each meal	A randomized crossover design compared two 8 h conditions: (1) Uninterrupted sitting (SIT) (2) Alternating between sitting and standing every 30 min (SIT-STAND).	Although sleepiness increased throughout the day in both groups, the SIT-STAND condition significantly attenuated this increase. However, this effect became non-significant when the final measurement of the day was removed from the analysis.
9	Kowalsky RJ et al. [32]	Participants consumed a single standardized meal, which was calculated to provide 30% of their daily caloric requirement (based on the Harris-Benedict Equation).	The study compared two 4 h conditions in a randomized order: (1) uninterrupted sitting (SIT); (2) sitting with hourly resistance exercise breaks (REX), which included exercises like chair stands and desk push-ups.	No statistically significant difference in reported sleepiness was found between the group that sat continuously and the group that performed hourly resistance exercise breaks.

## Data Availability

No new data were created or analyzed in this study. Data sharing is not applicable to this article as no datasets were generated or analyzed during the current study. All data analyzed are from publicly available articles that are cited in the text and listed in the references.

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
