# Peer review of "The Influence of Food Intake and Blood Glucose on Postprandial Sleepiness and Work Productivity: A Scoping Review"

_nutrients, 2025, doi:10.3390/nu17203217_

Round 1

Reviewer 1 Report

Comments and Suggestions for Authors

The objective of the authors is to investigate the relationship between postprandial sleepiness and blood glucose fluctuations, and to assess how these factors may influence workplace productivity. This approach is particularly relevant because it combines lifestyle, dietary, and physiological variables—areas that are often studied separately—offering a novel and comprehensive perspective that can guide future interventions aimed at improving employee health and performance. However, there are several issues that should be further discussed and clarified to strengthen the conclusions of the study.

  1. The introduction presents a well-structured overview of the economic and social context of workplace productivity, providing a strong conceptual foundation. However, the transition from macroeconomic factors (e.g., innovation, automation, wages) to the physiological analysis of diet- and glucose-related sleepiness is abrupt. The inclusion of a brief bridging explanation of how macro-level determinants influence individual mechanisms of sleep, diet, and glucose regulation would improve logical coherence and the overall flow of the argument.
  2. The manuscript mentions the inclusion of grey literature in the search strategy; however, the specific platforms (e.g., OpenGrey, ProQuest Dissertations) and criteria used for selecting these sources are not detailed. Providing this information would clarify how the quality and relevance of grey literature were ensured and would strengthen the transparency and reproducibility of the review process.
  3. In the Methods section, the authors indicate that studies involving participants with medical conditions were excluded, presumably to focus on a healthy working population and isolate the effects of diet, blood glucose fluctuations, and sleepiness. However, in the Discussion, some cited studies (e.g., Liviya et al. [24]) include participants with elevated BMI or obesity—conditions that, while not always classified as “diseases” per se, represent metabolic risk factors and are frequently considered chronic health conditions. This raises a potential inconsistency between the stated inclusion/exclusion criteria and the characteristics of the studies discussed, which should be clarified.
  4. Excluding participants with chronic physical or mental conditions may reduce potential bias, but it also limits the generalizability of the findings to real-world work environments, where such conditions are increasingly common. This point could be addressed in the limitations section.
  5. Table 1 provides a useful summary; however, it should be ensured that it precisely matches the inclusion and exclusion criteria detailed in Section 2.2 to avoid inconsistencies. For instance, “after-hours workers” are mentioned as excluded in the table, whereas this term does not appear explicitly in the main text.
  6. The exclusion of environments “where productivity is not emphasized” is an interesting criterion, but it seems subjective. It is unclear how the authors assessed whether productivity was indeed “emphasized.” For example, in certain academic or volunteer settings, productivity may be relevant even if it is not explicitly measured. Clarification on the operationalization of this criterion would strengthen methodological transparency.
  7. Most of the studies included small sample sizes, which limit statistical power. Additionally, few studies simultaneously assessed sleepiness, blood glucose, and productivity, making it difficult to establish causal relationships. These points should be explicitly addressed as limitations in the Discussion section.
  8. An important aspect that warrants consideration is the potential influence of sex-specific hormonal factors on the relationship between postprandial glucose fluctuations and daytime sleepiness. Evidence from previous studies suggests that, in women, menstrual cycle–related hormonal variations, the menopausal transition, and the use of hormonal contraceptives can modulate circadian rhythms, glucose metabolism, and sleep quality. Fluctuations in estrogen and progesterone have been reported to alter insulin sensitivity and sleep–wake regulation, which may contribute to differences in postprandial somnolence between men and women. Future research should aim to include sex-stratified analyses and, when possible, control for menstrual phase or menopausal status to better characterize these physiological interactions.
  9. The included studies provide valuable insights into the effects of light-intensity exercise on postprandial glucose fluctuations; nevertheless, there are notable limitations that challenge definitive interpretation. One key issue is the heterogeneity of dietary protocols: the meals used across studies varied substantially in energy content, macronutrient composition, and timing—ranging from meal-replacement beverages to mixed high-fat meals. Such dietary variability can critically influence postprandial glycemic responses, making direct comparisons between studies difficult and potentially confounding the observed effects of exercise interventions.
  10. Another important limitation is the variability in the type, frequency, and duration of exercise interventions across studies. For instance, Brocklebank et al. implemented two-minute walking breaks every 20 minutes, Wennberg et al. used three-minute walking bouts every 30 minutes, and Han et al. evaluated continuous pedaling. The lack of standardization in exercise intensity, duration, and modality hampers the identification of a clear dose–response relationship and complicates the comparison of outcomes across studies.
  11. As shown in Table 4, although both studies controlled meals through caloric and macronutrient standardization, the context of prolonged fasting and fixed meal times reduces external validity, as it does not reflect the typical eating patterns observed during a standard workday.
  12. In the manuscript, the authors acknowledge the scarcity of direct literature on workplace sleepiness and glucose fluctuations, which justifies the relevance of this scoping review. They also highlight technological advances, such as continuous glucose monitoring (CGM) and non-invasive measurements, which open new opportunities for applied research in occupational settings. However, potential confounders—such as chronotype, type of work (cognitive vs. physical), or caffeine consumption—are not discussed, even though these factors may significantly influence both sleepiness and glycemic responses.

Author Response

Thank you for reviewing the manuscript titled "Evaluating Sleepiness That Affects Work Productivity: A Scoping Review"(Manuscript ID: nutrients-3903274 - Major Revision). We have completed the revisions based on the reviewers' comments. Please review the final version.

Reviewer 1

1

The introduction presents a well-structured overview of the economic and social context of workplace productivity, providing a strong conceptual foundation. However, the transition from macroeconomic factors (e.g., innovation, automation, wages) to the physiological analysis of diet- and glucose-related sleepiness is abrupt. The inclusion of a brief bridging explanation of how macro-level determinants influence individual mechanisms of sleep, diet, and glucose regulation would improve logical coherence and the overall flow of the argument.

Thank you for your feedback. I understand that I need to ensure smoother transitions between sentences. Therefore, I added the following text at the beginning of the second paragraph in the Background section (line 67).

“While these organizational and environmental factors are significant, it is also crucial to focus on the productivity of individual workers. As automation handles more routine work, the value of human labor increasingly shifts to cognitive tasks where an individual's physiological state—governing factors like concentration and alertness—plays a decisive role in their performance [6]. Therefore, understanding the immediate factors that impact a worker's daily capacity, such as sleepiness, becomes a central issue for improving modern workplace productivity.”

2

The manuscript mentions the inclusion of grey literature in the search strategy; however, the specific platforms (e.g., OpenGrey, ProQuest Dissertations) and criteria used for selecting these sources are not detailed. Providing this information would clarify how the quality and relevance of grey literature were ensured and would strengthen the transparency and reproducibility of the review process.

Thank you for your feedback. I added the underlined text.

 “To complement the initial search, we conducted a thorough targeted search of the gray literature using Google Scholar and Web of Science to identify non-indexed studies, including unpublished trial data, articles or dissertations, and conference proceedings. The reference lists of relevant reviews and included studies were also manually searched for relevant studies that were not captured in the initial search; however, these were not included in the final list of reviewed studies”

3

In the Methods section, the authors indicate that studies involving participants with medical conditions were excluded, presumably to focus on a healthy working population and isolate the effects of diet, blood glucose fluctuations, and sleepiness. However, in the Discussion, some cited studies (e.g., Liviya et al. [24]) include participants with elevated BMI or obesity—conditions that, while not always classified as “diseases” per se, represent metabolic risk factors and are frequently considered chronic health conditions. This raises a potential inconsistency between the stated inclusion/exclusion criteria and the characteristics of the studies discussed, which should be clarified.

Thank you for pointing out this important point. We have revised Table 1, specifically the "Population/Excluded" section, as follows.

Participants included patients with diagnosed conditions such as diabetes, heart disease, and psychiatric disorders

(It also encompasses individuals who have risk factors for disease, like obesity, but have not yet been diagnose)

4

Excluding participants with chronic physical or mental conditions may reduce potential bias, but it also limits the generalizability of the findings to real-world work environments, where such conditions are increasingly common. This point could be addressed in the limitations section.

Your point is important. This review focused on individuals without pre-existing medical conditions. We hope that future reviews will include individuals with illnesses relevant to real-world work environments. We have revised Table 1 to clarify the inclusion criteria. Furthermore, we have outlined the limitations of this study in the discussion section.  Additionally, because the exclusion criteria for this review were based on "patients with diagnosed conditions," it's possible that workers with diagnosed illnesses in real-world work environments were inadvertently included, which could potentially lead to new insights.

5

Table 1 provides a useful summary; however, it should be ensured that it precisely matches the inclusion and exclusion criteria detailed in Section 2.2 to avoid inconsistencies. For instance, “after-hours workers” are mentioned as excluded in the table, whereas this term does not appear explicitly in the main text.

Thank you for your feedback. We have revised the wording regarding exclusions at the end of section 2.2 to align it with the table. Additionally, we have changed "after-hours workers" to "non-working hour employees" as the former term could be misleading.

“To exclude influences other than work and meals and to limit the study to actual workers, we applied the following exclusion criteria: (1) studies including patients with diabetes, heart disease, or other physical diseases; (2) studies including patients with mental illness; (3) studies that did not evaluate the relationship between food intake, sleepiness, and productivity; (4) studies involving students or those conducted in sports environments; and (5) studies that included data from non-working hours.

6

The exclusion of environments “where productivity is not emphasized” is an interesting criterion, but it seems subjective. It is unclear how the authors assessed whether productivity was indeed “emphasized.” For example, in certain academic or volunteer settings, productivity may be relevant even if it is not explicitly measured. Clarification on the operationalization of this criterion would strengthen methodological transparency.

Thank you for pointing out this important point. We have revised the context as follows.

Included

-Changes in sleepiness after meals

-Changes in productivity metrics (objective or subjective) linked to post-meal states

Excluded

-Sleepiness that is unrelated to food intake

-Productivity unrelated to drowsiness

7

Most of the studies included small sample sizes, which limit statistical power. Additionally, few studies simultaneously assessed sleepiness, blood glucose, and productivity, making it difficult to establish causal relationships. These points should be explicitly addressed as limitations in the Discussion section.

Thank you for pointing out this important point. We have outlined the limitations of this study in the discussion section.
“First, a significant limitation within the existing literature is that most of the included studies had small sample sizes, which restricts their statistical power and the generalizability of the findings. Second, very few studies simultaneously assessed the key variables of sleepiness, blood glucose, and productivity. This scarcity of comprehensive research makes it difficult to establish clear causal relationships and represents a major gap in the field.”

8

An important aspect that warrants consideration is the potential influence of sex-specific hormonal factors on the relationship between postprandial glucose fluctuations and daytime sleepiness. Evidence from previous studies suggests that, in women, menstrual cycle–related hormonal variations, the menopausal transition, and the use of hormonal contraceptives can modulate circadian rhythms, glucose metabolism, and sleep quality. Fluctuations in estrogen and progesterone have been reported to alter insulin sensitivity and sleep–wake regulation, which may contribute to differences in postprandial somnolence between men and women. Future research should aim to include sex-stratified analyses and, when possible, control for menstrual phase or menopausal status to better characterize these physiological interactions.

Thank you for pointing out this important point. We have outlined the limitations of this study in the discussion section.

“Third, the potential influence of sex-specific hormonal factors was not considered in the reviewed literature. Hormonal fluctuations related to the menstrual cycle or menopause can modulate glucose metabolism and sleep quality, yet these variables are rarely controlled for. Future research should incorporate sex-stratified analyses to clarify these interactions“

9

The included studies provide valuable insights into the effects of light-intensity exercise on postprandial glucose fluctuations; nevertheless, there are notable limitations that challenge definitive interpretation. One key issue is the heterogeneity of dietary protocols: the meals used across studies varied substantially in energy content, macronutrient composition, and timing—ranging from meal-replacement beverages to mixed high-fat meals. Such dietary variability can critically influence postprandial glycemic responses, making direct comparisons between studies difficult and potentially confounding the observed effects of exercise interventions.

Thank you for pointing out this important point. We added an explanation after the citation of paper 27 in the Discussion section.

Although some studies (e.g., Brocklebank et al. [27]) have reported changes in postprandial glycemic responses with light-intensity activity, interpreting these discrepant findings is challenging due to methodological inconsistencies across studies. A primary issue is the variability in exercise interventions; for instance, protocols ranged from two-minute walking breaks every 20 minutes [27] to continuous pedaling, making it difficult to establish a clear dose-response relationship. Compounding this, the dietary protocols were also highly heterogeneous, with meals varying substantially in energy content, macronutrient composition, and timing. This lack of standardization in both exercise and dietary controls complicates direct comparisons and may confound the observed outcomes.

10

Another important limitation is the variability in the type, frequency, and duration of exercise interventions across studies. For instance, Brocklebank et al. implemented two-minute walking breaks every 20 minutes, Wennberg et al. used three-minute walking bouts every 30 minutes, and Han et al. evaluated continuous pedaling. The lack of standardization in exercise intensity, duration, and modality hampers the identification of a clear dose–response relationship and complicates the comparison of outcomes across studies.

Thank you for your feedback. This is included in the answer to comment 9.

11

As shown in Table 4, although both studies controlled meals through caloric and macronutrient standardization, the context of prolonged fasting and fixed meal times reduces external validity, as it does not reflect the typical eating patterns observed during a standard workday.

Thank you for your feedback. We added the following sentence before Table 5 (Table 4 before modification):

“Although the included studies aimed to simulate working hours, their experimental protocols deviated from typical workplace conditions. Specifically, the prescribed fasting periods and exercise interventions do not reflect standard practices in a real-world occupational setting, which may limit the generalizability of the findings.”

12

In the manuscript, the authors acknowledge the scarcity of direct literature on workplace sleepiness and glucose fluctuations, which justifies the relevance of this scoping review. They also highlight technological advances, such as continuous glucose monitoring (CGM) and non-invasive measurements, which open new opportunities for applied research in occupational settings. However, potential confounders—such as chronotype, type of work (cognitive vs. physical), or caffeine consumption—are not discussed, even though these factors may significantly influence both sleepiness and glycemic responses.

Thank you for your important point. It's very helpful. I've added the following as the second paragraph from the bottom of the Discussion section.

“Building on recent technological advances, future applied workplace research should incorporate the assessment of key potential confounders. Individual chronotype (e.g., morningness–eveningness), job characteristics (cognitive vs. physical workload), and caffeine consumption each influence both sleepiness and glycemic responses. For chronotype, individuals with stronger eveningness or greater social jetlag tend to exhibit lower daytime alertness, higher fasting glucose and HbA1c, and an elevated risk of type 2 diabetes [43]. Regarding job characteristics, prolonged cognitively demanding, predominantly sedentary work increases mental fatigue and sleepiness. In contrast, interrupting sedentary time with short bouts of light walking reduces postprandial glycemic and insulinemic responses, as shown by intervention trials and reviews. Moreover, physically active tasks enhance skeletal muscle glucose uptake and improve insulin sensitivity; for example, aerobic exercise augments mitochondrial function and insulin receptor activity, and with continued practice, lowers fasting glucose and HbA1c levels. By contrast, seated cognitive work expends little energy and predisposes workers to postprandial hyperglycemia. Experimentally, brief stair‐climbing after meals significantly attenuates the rise in postprandial glucose, indicating that modest activity breaks blunt glycemic spikes and supporting the utility of exercise interventions within cognitively oriented, sedentary workplaces [44]. These observations are consistent with, and extend, prior findings by Wennberg et al. [26] and Brocklebank et al. [27]. Caffeine use is also common among workers (e.g., coffee, energy drinks). Through central adenosine-receptor antagonism, caffeine increases wakefulness and reduces sleepiness; however, it acutely decreases insulin sensitivity and can transiently shift glycemia upward [45].

 Taken together, these mechanisms and empirical findings suggest that chronotype, job characteristics, and caffeine intake can serve as significant confounders when assessing postprandial sleepiness and glycemic trajectories. Appropriately controlling these variables is essential for isolating the specific impact of dietary factors on productivity and for designing more targeted and effective workplace interventions.”

Reviewer 2 Report

Comments and Suggestions for Authors

Review of “Evaluating sleepiness that affects work productivity: a scoping review”

-Title: I suggest including information that indicates the main methods are tied to “food intake” and “blood glucose levels”, as it initially did not seem to be a paper that would fit the scope of Nutrients.

-Abstract, objective: “caused by fluctuations in food” would it be “food intake”?

-Abstract, methods: please, put all the databases that were searched (and not “such as”).

-Abstract, line 24: It seems that there is a missing word before “extracted”.

-Main text: Please, move the prisma flow diagram to the results section, as the checklist suggests.

-Table 2: Please, create a column with the “type of study”. It is difficult to determine whether the studies were controlled trials (randomized or not) or observational studies. Maybe there is some cue to it in the “effects of eating” but this nomenclature is awkward.

-Table 2: The column “participants' characteristics” is also awkward, since it brings several information such as the sample size. Authors could reframe Table 2, ensuring that each column represents only one piece of information (and not multiple pieces). And maybe move the last column “summary” to a new table where the results of each paper could be better explored and summarized (since this table is already too narrow). Columns “authors” and “year of publication” could be merged. Column Age displays “SD = XX” and also includes “+-”, which can be confusing.

-Table 2: Please, be consistent if reporting the male/female ratio by % or by actual numbers.

-Table 2, study 2: Please, define what “young” actually means. Show the actual age range.

-Table 3 and 4, headings: The headings beginning with the verb “compare” are awkward. Perhaps something like “characteristics of the studies that….” would be more suitable.

Results: could the authors use any kind of methodological quality rating on the included studies? JBI scales could be helpful. It is essential to assess the quality of the included studies in many scoping/systematic reviews and critically appraise the evidence in light of this quality.

-Line 300-301: There is an error in the citation.

Author Response

Thank you for reviewing the manuscript titled "Evaluating Sleepiness That Affects Work Productivity: A Scoping Review"(Manuscript ID: nutrients-3903274 - Major Revision). We have completed the revisions based on the reviewers' comments. Please review the final version.

Reviewer 2

1

-Title: I suggest including information that indicates the main methods are tied to “food intake” and “blood glucose levels”, as it initially did not seem to be a paper that would fit the scope of Nutrients.

As you suggested, to make it more clear, we have changed the title to: “The Influence of Food Intake and Blood Glucose on Postprandial Sleepiness and Work Productivity: A Scoping Review”

2

Abstract, objective: “caused by fluctuations in food” would it be “food intake”?

Thank you for pointing out this. As you pointed out, “food intake” is communicated correctly. I've corrected it.

3

-Abstract, methods: please, put all the databases that were searched (and not “such as”).

We only used PubMed, and Web of Science so "such as" is not required, so I deleted it.

4

Abstract, line 24: It seems that there is a missing word before “extracted”.

Thank you for pointing out this. It's our miswriting. Fixed as "The extracted literature also included..."

5

Main text: Please, move the prisma flow diagram to the results section, as the checklist suggests

As mentioned in the comments, I moved to Result.

6

Table 2: Please, create a column with the “type of study”. It is difficult to determine whether the studies were controlled trials (randomized or not) or observational studies. Maybe there is some cue to it in the “effects of eating” but this nomenclature is awkward

The following revisions have been completed

- Added Type of study

- "effects of eating" is fixed to "Dietary Control."

7

Table 2: The column “participants' characteristics” is also awkward, since it brings several information such as the sample size. Authors could reframe Table 2, ensuring that each column represents only one piece of information (and not multiple pieces). And maybe move the last column “summary” to a new table where the results of each paper could be better explored and summarized (since this table is already too narrow). Columns “authors” and “year of publication” could be merged. Column Age displays “SD = XX” and also includes “+-”, which can be confusing.

The following revisions have been completed:

- The "Author/Publication Year" columns have been merged.

- Tables 2 and 3 have been reorganized, with a summary added to Table 3.

- The section on "Participant characteristics" was removed because its content was too abstract.

- A column for "Sample size" only has been added.

- The age range is now standardized using ± notation.

8

Table 2: Please, be consistent if reporting the male/female ratio by % or by actual numbers

The number of female/male participants is now explicitly stated (since the sample sizes were small, we judged this to be more appropriate than using percentages).

9

Table 2, study 2: Please, define what “young” actually means. Show the actual age range

For Study 2: Regarding "young," we reviewed the original paper and added the specific age range. Thank you for your feedback.

10

Table 3 and 4, headings: The headings beginning with the verb “compare” are awkward

Based on your comments received, we have made the following revisions:

Table 4. Characteristics of studies that examined the relationship between light exercise and blood glucose levels.

Table 5. Characteristics of studies that examined the relationship between light exercise and sleepiness.

11

-Line 300-301: There is an error in the citation.

Thank you for your comment. I have re-checked the paper by Madvari RF et al. (Citation 40) and confirmed that it states: “Drivers are also exposed to street lighting, which affects mood and sleepiness”. However, I understand your comment that this wording might not be clear enough, so I have revised it as follows: “Drivers can experience certain perceptual errors, such as sleepiness, due to insufficient street lighting.”

1

Data Availability Statements are required for all articles that are published with MDPI. During the peer review and editorial decision process, authors can be asked to share existing datasets or raw data that have been analyzed in the manuscript, and may be asked to indicate whether they will be made available to other researchers following publication. The authors will also be asked for the details of any existing datasets that have been analyzed in the manuscript. We hope you could complete the Data Availability Statements in the manuscript backmatter, for more details, please see: https://www.mdpi.com/ethics#_bookmark21

Thank you for pointing that out.We added the following text before the "References" section:

“Data Availability Statement: No new data were created or analyzed in this study. Data sharing is not applicable to this article as no datasets were generated or analyzed during the current study. All data analyzed are from publicly available articles that are cited in the text and listed in the references.”

2

Additionally, we found the duplication rate of your paper is high and has some duplication in whole paragraphs. The duplication rate with a single article should be less than 30% and should not have duplication in whole paragraphs. So, we attached the duplication report here and hope you could revise it accordingly, together with the review reports from reviewers.

Thank you for your important point. We have revised the sections in the Methods and Results that quote text from the original research papers.

We also added content to Table 3.4.5.

*Note that the table numbers are different from the original paper, as Table 2 has been split into Table 2 and 3.

Round 2

Reviewer 1 Report

Comments and Suggestions for Authors

After carefully reviewing the revised version of the manuscript, I consider that the authors have satisfactorily addressed most of the proposed comments. My specific remarks are as follows:

  1. The added text effectively integrates macro-level factors (automation, work organization) with micro-level factors (physiological state, alertness, and sleepiness). However, the connection with diet and glucose could be slightly strengthened, as the term “physiological state” is mentioned in a general sense. A sentence such as:

“…among which sleep regulation, diet, and glucose metabolism play a decisive role in cognitive performance” would further reinforce the transition.

2.The inclusion of gray literature through Google Scholar is acknowledged, which is appropriate as this platform indexes theses, reports, and other materials outside traditional academic publications. However, Web of Science primarily focuses on peer-reviewed publications and academic journal articles, rather than gray literature. It would be advisable to acknowledge this, although both databases were used, they are not the most suitable sources for gray literature.

3.I appreciate the modification made to Table 1. This improves internal consistency and clarity regarding the inclusion/exclusion criteria.

4.The response regarding the exclusion of participants with chronic conditions is satisfactory: it addresses the comment reasonably and enhances transparency. Additionally, the acknowledgment that “some workers with diagnosed conditions may have been inadvertently included” reflects commendable methodological honesty.

5.The consistency between Table 1 and the text in Section 2.2 has been fully and appropriately addressed, and the terminology has been correctly clarified.

6.The specific criteria used to include or exclude studies are now clearer, focusing on the direct relationship between food intake, sleepiness, and productivity. This reduces the subjectivity of the original criterion, improving transparency and reproducibility.

7.The added section on the methodological limitations of the included studies is appropriate. It could be further strengthened by briefly indicating how these limitations affect the interpretation of the review’s findings, so that conclusions can be drawn with due caution.

8.The response regarding sex-specific hormonal factors appropriately recognizes their importance in the regulation of sleepiness and glucose metabolism.

9 and 10. I appreciate the inclusion in the Discussion of limitations related to the heterogeneity of diet and exercise protocols. A brief comment on the impact of these inconsistencies on the review’s conclusions, rather than solely describing the issue, could further enhance clarity.

11.I thank the authors for considering the limited external validity due to prolonged fasting periods and fixed meal timings; this addition strengthens the interpretation of the findings.

12. The response regarding potential confounding factors (chronotype, type of work, and caffeine intake) fully addresses the observation. The discussion is extensive, detailed, and evidence-based, clearly explaining the physiological mechanisms and their relevance for interpreting sleepiness and glycemic responses in occupational settings.